# Secondary Metabolite Variation and Bioactivities of Two Marine *Aspergillus* Strains in Static Co-Culture Investigated by Molecular Network Analysis and Multiple Database Mining Based on LC-PDA-MS/MS

**DOI:** 10.3390/antibiotics11040513

**Published:** 2022-04-12

**Authors:** Yuan Wang, Evgenia Glukhov, Yifan He, Yayue Liu, Longjian Zhou, Xiaoxiang Ma, Xueqiong Hu, Pengzhi Hong, William H. Gerwick, Yi Zhang

**Affiliations:** 1College of Food Science and Technology, Guangdong Ocean University, Guangdong Provincial Key Laboratory of Aquatic Product Processing and Safety, Guangdong Province Engineering Laboratory for Marine Biological Products, Guangdong Provincial Engineering Technology Research Center of Seafood, Shenzhen Institute of Guangdong Ocean University, Zhanjiang Municipal Key Laboratory of Marine Drugs and Nutrition for Brain Health, Research Institute for Marine Drugs and Nutrition, Guangdong Ocean University, Zhanjiang 524088, China; wangyuan5614@163.com (Y.W.); liuyayue@gdou.edu.cn (Y.L.); zhoulongjian@gdou.edu.cn (L.Z.); maxiaoxiang97@outlook.com (X.M.); hxwz247@163.com (X.H.); hpzgd13902501729@126.com (P.H.); 2Collaborative Innovation Center of Seafood Deep Processing, Dalian Polytechnic University, Dalian 116034, China; 3Center for Marine Biotechnology and Biomedicine, Scripps Institution of Oceanography, and the Skaggs School of Pharmacy and Pharmaceutical Sciences, University of California, San Diego, La Jolla, CA 92093, USA; eglukhov@ucsd.edu (E.G.); roderick98hyf@gmail.com (Y.H.); wgerwick@health.ucsd.edu (W.H.G.)

**Keywords:** *Aspergillus terreus*, *Aspergillus unguis*, co-culture, antimicrobial activity, LC-PDA-MS/MS, molecular network, database mining

## Abstract

Co-culture is known as an efficient way to explore the metabolic potential of fungal strains for new antibiotics and other therapeutic agents that could counter emerging health issues. To study the effect of co-culture on the secondary metabolites and bioactivities of two marine strains, *Aspergillus terreus* C23-3 and *Aspergillus. unguis* DLEP2008001, they were co-cultured in live or inactivated forms successively or simultaneously. The mycelial morphology and high-performance thin layer chromatography (HPTLC) including bioautography of the fermentation extracts were recorded. Furthermore, the agar cup-plate method was used to compare the antimicrobial activity of the extracts. Based on the above, liquid chromatography-photodiode array-tandem mass spectrometry (LC-PDA-MS/MS) together with Global Natural Products Social molecular networking (GNPS) and multiple natural products database mining were used to further analyze their secondary metabolite variations. The comprehensive results showed the following trends: (1) The strain first inoculated will strongly inhibit the growth and metabolism of the latter inoculated one; (2) Autoclaved *A. unguis* exerted a strong inducing effect on later inoculated *A. terreus*, while the autoclaved *A. terreus* showed high stability of its metabolites and still potently suppressed the growth and metabolism of *A. unguis*; (3) When the two strains are inoculated simultaneously, they both grow and produce metabolites; however, the *A. terreus* seemed to be more strongly induced by live *A. unguis* and this inducing effect surpassed that of the autoclaved *A. unguis*. Under some of the conditions, the extracts showed higher antimicrobial activity than the axenic cultures. Totally, *A. unguis* was negative in response but potent in stimulating its rival while *A. terreus* had the opposite effect. Fifteen MS detectable and/or UV active peaks showed different yields in co-cultures vs. the corresponding axenic culture. GNPS analysis assisted by multiple natural products databases mining (PubChem, Dictionary of Natural Products, NPASS, etc.) gave reasonable annotations for some of these peaks, including antimicrobial compounds such as unguisin A, lovastatin, and nidulin. However, some of the peaks were correlated with antagonistic properties and remain as possible novel compounds without mass or UV matching hits from any database. It is intriguing that the two strains both synthesize chemical ‘weapons’ for antagonism, and that these are upregulated when needed in competitive co-culture environment. At the same time, compounds not useful in this antagonistic setting are downregulated in their expression. Some of the natural products produced during antagonism are unknown chlorinated metabolites and deserve further study for their antimicrobial properties. In summary, this study disclosed the different responses of two *Aspergillus* strains in co-culture, revealed their metabolic variation, and displayed new opportunities for antibiotic discovery.

## 1. Introduction

The ocean supports an amazing variety of marine life and is a crucial part of the biosphere. Marine organisms, including microbes, have developed complex metabolic mechanisms to adapt themselves to the unique environment of high salinity, high pressure, low oxygen, and oligotrophy. As a result, secondary metabolites (SMs) with novel structures and rich activities are produced and provide a rich source of drug lead compounds [1]. Although medical sciences have made significant progress, infectious diseases caused by bacteria, fungi, and viruses still pose a substantial threat to public health. Due to the development of antibiotic resistance, finding new antibiotics remains an essential task for scientists worldwide. Nevertheless, most of the current antibacterial agents derived from natural products were isolated from terrestrial sources, while marine organisms are still primarily untapped resources for new biologically active natural products, and especially antibiotics [2].

Previous studies have shown that fungi contain many diverse biosynthetic gene clusters that encode for secondary metabolites, but under artificial culture conditions in the laboratory, most fungal functional gene clusters are silent (i.e., not expressed). The methods for enriching the diversity of expressed fungal metabolites include changing the composition of the medium, changing the environmental conditions, adding epigenetic modifiers, and co-cultivating with other fungi or bacteria [3,4,5]. Specifically, the co-cultivation of microorganisms from different sources often creates competition and antagonism. To compete for the limited natural resources in such an environment, or for living space or to maintain information transmission between species, the microbes often produce secondary metabolites that are not produced when cultured separately [6].

Mass spectrometry (MS)-based metabolomics is increasingly playing an important role in efficient natural products studies. These approaches enable the accurate offline and online comparison of constituent differences among samples including big data samples. However, these data sets are often too large for manual analysis as more than 1000 MS/MS spectra can be collected from just one extract sample. Global Natural Products Social molecular networking (GNPS) is a data-driven open platform for the storage, analysis, and dissemination of MS/MS spectra. It provides the ability to visualize data sets from different users and compare these with all publicly available reference spectra to annotate known molecules and discover putative analogs [7]. For example, Oppong-Danquah. et al. used a GNPS molecular networking-based screening method to annotate metabolites with crop protection activity in co-cultures of several marine fungi, significantly improving the efficiency of discovery and identification of trace novel natural products [8]. GNPS is a continuously developing platform for accurate dereplication and annotation tasks, and thus is best complemented with other approaches at the present time.

In preliminary studies, our laboratory obtained a marine *Aspergillus terreus* strain C23-3 from a coral collected in Xuwen Natural Reserve of South China Sea and a marine *A. unguis* strain DLEP2008001 from a seaweed collected at the intertidal zone of Dalian City by the Yellow Sea of Northern China. Intriguingly, our previous research and the reports from other groups indicated that both these two species/strains can produce potent antibiotics as well as other bioactive compounds. For example, *A. terreus* produces butyrolactones showing antibacterial, antitumor, antioxidant, antiviral, enzymes (glucosidase, glucuronidase, and cyclin-dependent protein kinase 5) inhibitory, neuroprotective, anti(-neuro-)inflammatory, and axonal growth promoting activities [9,10,11,12,13,14], as well as lipid lowering lovastatins and acetylcholinesterase (AChE) inhibitory territrems [15,16]. As for *A. unguis*, it was known to produce halogenated and non-halogenated depsidones which were reported to possess antibacterial, antifungal, brine shrimp larvacidal, enzyme inhibitory (AChE and aromatase), diphenyl-picryl hydrazyl (DPPH) free radical scavenging, and neuroprotective activities [17,18,19,20,21,22,23,24,25].

Because both strains are producers of antibiotics and neuroactive agents, it was appealing to investigate the effect of co-culture conditions on expression of their secondary metabolites, as this might result in the discovery of new antibiotics or anti-neurodegenerative agents. In this paper, we investigated the high-performance thin layer chromatographic (HPTLC) profiles, bioactivities, LC-MS/MS based GNPS molecular networking, and multiple natural product database mining of secondary metabolites deriving from co-culturing of these two marine fungi.

## 2. Results

### 2.1. Morphological Comparison

The two strains *A. unguis* and *A. terreus* were statically cultivated for a total of 28 days (in one stage or two stages) in seawater potato sucrose broth under 7 experimental condition groups, including: G1) *A. unguis* axenically for 28 days (abbreviated as axU); G2) *A. terreus* axenically for 28 days (axT); G3) *A. unguis* 7-day culture-inactivation + live *A. terreus* for the following 21 days (iacU-livT); G4) Live *A. unguis* 7-day culture + live *A. terreus* for the following 21 days (livU-livT); G5) live *A. unguis* and live *A. terreus* inoculated simultaneously (livU/livT) and co-cultivated for 28 days; G6) *A. terreus* 7-day culture-inactivation + live *A. unguis* for the following 21 days (iacT-livU); G7) Live *A. terreus* 7-day culture + live *A. unguis* for the following 21 days (livT-livU) (see details of the culture experiments in Section 4.2.1).

When *A. unguis* grows axenically, it initially forms scattered bright yellow colonies and then merges into a dark brown mycoderm (Figure 1G1). When *A. terreus* grows alone, it initially forms white colonies and then expands to form an off-white to brown mycoderm (Figure 1G2). If *A. terreus* is inoculated on the autoclaved 7-day culture of *A. unguis*, it still grows but just as small scattered off-white colonies on the dead mycoderm of *A. terreus* without forming its own continuous mycoderm (Figure 1G3). If *A. terreus* is inoculated on the live 7-day culture of *A. unguis*, no obvious growth of *A. terreus* is observed (Figure 1G4). When *A. unguis* and *A. terreus* are inoculated simultaneously, two types of mycoderms are observed: dark brown and off-white (Figure 1G5). When *A. unguis* is inoculated on the autoclaved 7-day culture of *A. terreus*, it does not grow as well as the axenic *A. unguis* culture but still forms its own complete mycoderm on the surface of the dead mycoderm of *A. terreus* (Figure 1G6). However, if *A. unguis* is inoculated on the live 7-day culture of *A. terreus*, the newcomers’ growth is not obvious (Figure 1G7). Comparing the different culture experiments, it was found that the early inoculated fungus (even if autoclaved) will inhibit the growth of the late inoculated one, and the live fungus showed stronger inhibition than the inactivated one.

### 2.2. Comparison of HPTLC Fingerprints

The profiles of secondary metabolites including their antioxidant as well as anti-AChE constituents were demonstrated using HPTLC images that were observed under 254 nm and 365 nm, colored by anisaldehyde reagent and potassium ferricyanide-ferric chloride (PFFC) reagent, and revealed by DPPH free radical scavenging and AChE inhibitory bioautographies, respectively. These HPTLC images revealed the variation between axenic cultures and co-cultures in different ways, including some dramatical changes.

The UV images (under 254 nm and 365 nm) clearly showed rich secondary metabolic profiles from the cultures. In the profile of axenic *A. unguis* (lane 1 in Figure 2A under 254 nm), the big dark spot with Rf value of 0.70 was judged to be comprised of depsidones according to our previous study on this strain [18]. For axenic *A. terreus*, the dark spot with Rf value of 0.28 was recognized from previous work as butyrolactone I [11]. These annotations were also supported by the LC-PDA-MS/MS analysis as described below in Section 2.4.

In the experiment G3 (iacU-livT), some *A. terreus* metabolites disappeared including the spots at Rf 0.43 (orange fluorescence), Rf 0.38 (red fluorescence), and Rf 0.17–0.30 (dark blue, including butyrolactone I). However, other putative *A. terreus* metabolites were enhanced in their production, including white fluorescent spots at Rf 0.84 and Rf 0.68. Moreover, several new constituents appeared in this co-culture including fluorescent spots at Rf 0.76 (light orange), Rf 0.58 (white), and Rf 0.20 (white) as marked with the yellow arrows (Figure 2B under 365 nm). Remarkably, the typical *A. unguis* depsidone metabolites still appeared in this co-culture (Figure 2A under 254 nm), indicating their thermostability against autoclaving.

In the experiment G4 (livU-livT, inoculated in tandem), the UV images were basically the same as those of the axenic *A. unguis*, indicating that the later inoculated *A. terreus* was extremely suppressed in its growth and production of metabolites. This is consistent with the morphological observation described above.

In experiment G5 (livU/livT, inoculated simultaneously), the UV images greatly differed from those of G3 and G4, but closely resembled those of G2 (the axenic *A. terreus*). However, some *A. terreus* products were produced in lower yields such as the spot with Rf 0.38 (red fluorescence), or even vanished such as the one with Rf 0.43 (orange fluorescence). In contrast, some spots, like the white fluorescent spots with Rf 0.84, 0.67, 0.32, and 0.08, were significantly enhanced to a much higher extent than in co-culture G3. As for *A. unguis* metabolites, only a small quantity of depsidone metabolites (Rf 0.67) was observed under 254 nm. This situation agreed with the growth advantage of *A. terreus* vs. *A. unguis* in Figure 1G5.

In the experiment G6 (iacT-livU), the UV image under 365 nm was similar to that of axenic *A. terreus*, suggesting that the *A. terreus* metabolites were quite thermotolerant against autoclaving and were stable over a three-week period. The depsidones of *A. unguis* were present but with much lower yield compared to the axenic *A. unguis* culture. Therefore, even the autoclaved *A. terreus* can remarkably inhibit the growth or metabolism of *A. unguis*.

Likewise, in experiment G7 (livT-livU, inoculated in tandem), the UV image displayed almost identical features with the axenic *A. terreus* cultures; however, the depsidones from *A. unguis* could barely be observed.

The results of anisaldehyde and PFFC colorization (Figure 2C,D) were consistent with the UV findings. Additionally, they revealed the production of new metabolite during the co-cultivation experiments. For example, the following were new compounds: blue gray spot at Rf 0.58 in experiment G3, gray spot at Rf 0.82 in G4 (both with anisaldehyde detection), brown spot at Rf 0.27 and brownish spot at Rf 0.40 (both with PFFC detection).

The two bioautographies revealed the antioxidant and anti-AChE constituents in the different culture experiments (Figure 2E,F). The depsidones showed antioxidant and anti-AChE activities, while butyrolactone I showed antioxidant activity. Their variation in amounts (Figure 2A) were partially reflected in the changes of the bioactive spots in these bioautography experiments. Nevertheless, some minute new anti-AChE spots (Rf 0.32 and 0.36) were also observed in co-culture G3, and differences in highly polar constituents remaining at the point of application cannot be excluded because the mobile phase (chloroform:methanol = 20:1 (*v*/*v*)) was not polar enough to mobilize them in the chromatogram.

Generally, the HPTLC suggested the following trends: (1) the strain firstly inoculated will strongly inhibit the growth and metabolism of the later inoculated one, (2) the autoclaved *A. unguis* exerted a strong inducing effect on the later inoculated *A. terreus*, while the autoclaved *A. terreus* showed high stability of its metabolites and still potently suppressed the growth and metabolism of *A. unguis*, (3) when the two strain were inoculated simultaneously, they both grew and produced metabolites. However, the *A. terreus* seemed to be more strongly induced by live *A. unguis* and this inducing effect surpassed that of the autoclaved *A. unguis*. Finally, *A. unguis* was negative in response and agonism but potent in stimulating its rival while *A. terreus* had the opposite effect.

### 2.3. Antimicrobial Activity

Based on the above preliminary co-cultural product HPTLC analysis, the antimicrobial activities of the extracts were further tested against several indicator strains, including Methicillin-resistant *Staphylococcus aureus* (MRSA), *Bacillus subtilis*, *Pseudomonas aeruginosa*, *Vibro parahemolyticus*, *V. alginolyticus*, *Shewanella putrefaciens*, *Yersinia pseudotuberculosis*, and *Candida albicans*. The results are shown in Table 1 and the representative photos below in Figure 3.

In these experiments, axenic *A. unguis* extracts exhibited much stronger broad-spectrum antimicrobial activity against the selected indicator strains than axenic *A. terreus* extracts. However, as revealed by morphological evaluation and HPTLC fingerprints, the *A. terreus* possessed strong inhibition against *A. unguis*, even by its autoclaved medium. This antifungal activity, though, may be specific to *A. unguis*, since *C. albicans* showed low sensitivity to the axenic *A. terreus* products.

When *A. unguis* grew well and fully, such as in the livU-livT (G4) experiment, the overall activity of the co-culture reached comparable levels of axU (G1). However, it was noticed that its activities against *B. subtilis* and *P. aeruginosa* were remarkably higher than those of axU (G1), which was possibly related to the higher depsidone metabolites yield under this condition as shown by HPTLC.

While *A. unguis* was inactivated by autoclaving after the first week or grew weakly when co-inoculated or late inoculated, with low yields of depsidones, the antimicrobial spectra and the inhibition potency of the co-cultural products would be generally weakened, as depicted in the results of iacU-livT(G3), livU/livT(G5), iacT-livU(G6), and livT-livU(G7). Nevertheless, the anti-*B. subtilis* and anti-*S. putrefaciens* activities of livU/livT(G5) were enhanced by co-culture, and the anti-*C. albican* activity of iacU-livT(G3) also became higher than axU (G1), suggesting that new antimicrobial substances may be produced in co-cultures. Taking into account, too, the remarkable increase in total extract amounts, the extent of the activity enhancement is even more dramatic because the activities reported in Table 1 are the results from identical sample concentrations. Considering that the strong inducing effect of *A. unguis* towards *A. terreus* and the antifungal potential of *A. terreus* (against *A. unguis* in co-cultures), the antifungal activity of iacU-livT(G3) was possibly from the induced products of *A. terreus*.

### 2.4. Metabolits Profile Comparison by LC-PDA-MS/MS and Multiple Database Mining

To further investigate the metabolic profile variation occurring as a result of co-cultivation, and to annotate putative antimicrobial compounds and their derivatives, LC-PDA-MS/MS analysis and GNPS molecular network-based analyses were performed for the extracts of the seven experimental setups. As a supplement to GNPS automatic metabolite annotation, manual searching was also performed to find possible candidates for the compounds showing dramatic changes in yield. This latter analysis used several open accessible natural product databases including PubChem, Dictionary of Natural Products (DNP), NPASS, Natural Product Atlas, and Nmrdata (WeiPu). The deduced molecular weights, isotopic patterns (for chlorinated metabolites), UV features, and taxon information (mainly within the Genus of *Aspergillus* and expanded to the kingdom of fungi when necessary) were queried in these database searches.

In general, the LC-UV profiles (under 280 and 360–370 nm) together with the LC-MS BPC profiles (under both positive and negative MS modes) (Figure 4, Figure 5, Figure 6 and Figure 7) show the similar trends in fungal metabolite production in co-cultures as those observed by HPTLC. Especially, the 360–370 nm UV monitoring (although without fluorescent detection) showed rich upregulated peaks from *A. terreus* (G5 vs. G3) as displayed by the fluorescent components in HPTLC results but not by MS monitoring. These observations suggest that live *A. unguis* had a stronger inducing ability to *A. terreus* than the autoclaved one. These results also suggestion that the fluorescent substances may be not readily ionizable, or that they are present in very low quantity.

Totally, 15 MS detectable main peaks were chosen based on their significant variation in yield compared with the corresponding axenic cultures (Table 2) and submitted to annotation by GNPS and the multiple database mining approach described above. Their complete information is summarized in Table 3; the peaks are marked in Figure 4, Figure 5, Figure 6 and Figure 7 and the selected annotated structures are presented in Figure 8 and partially by GNPS in Figure 9 and Figure 10. By comparison of the axenic and co-culture LC profiles, peaks 1, 2, and 5–13 were assigned to be *A. terreus* metabolites, and peaks 2–4, 14, and 15 were assigned to *A. unguis* metabolites. Specially, peak 2 was produced by both strains.

Compared with the axenic *A. terreus*, most of its metabolites except peak 2 were remarkably decreased in yield when *A. terreus* was inoculated one week after *A. unguis* (i.e., groups G3: iacU-livT and G4: livU-livT). Moreover, live *A. unguis* showed a stronger suppressive effect than the autoclaved material, indicating the inhibition of *A. unguis* metabolites towards *A. terreus* (Table 2).

Peak 7 was annotated to be the typical *A. terreus* metabolite lovastatin by MS/MS similarity to GNPS records. Its UV features showed a difference to related literature values [15,16] which was possibly caused by unknown impurities in the peak. Peak 8 appeared in the same cluster in the GNPS molecular network (the node with average *m/z* 427.256 for peak 7 and the node with average *m/z* 459.273 for peak 8 in Figure 9A) and was therefore also proposed to be a statins. Peaks 7 and 8 were dramatically upregulated in G5 and G7 (when *A. terreus* was inoculated simultaneously with or prior to *A. unguis*, respectively), indicating that they were employed in antagonistic responses to *A. unguis*. This phenomenon is also consistent with the antifungal activity reported for statin natural products [26,27]. In G6 (iacT-livU), these two compounds remained in high concentration even though the 1-week *A. terreus* culture was autoclaved, indicating that they were produced in the early stage of growth and possessed good stability to persist in their inhibition to *A. unguis*. This was indicated by the weakened mycelial growth and downregulated metabolites observed in HPTLC and LC profiles of group G6 *A. unguis*.

Four peaks (1, 5, 10, and 11) all increased in G7 (livT-livU) but were reduced in the simultaneously inoculated G5 experiment (livT/livU). Among them, peaks 5 and 11 were also increased in group G6 (iacT-livU) but to a lesser extent than G7, but peaks 1 and 10 were decreased in G6. Their varied production levels suggest that all four compounds may contribute to the competitive success of pre-inoculated *A. terreus* growth in co-cultures; however, their potency may be too low to be effective in the spontaneous inoculation co-culture G5. Peaks 5 and 11 were deduced to be produced in the early stage of growth and stable, whereas peaks 1 and 10 may be not stable or are synthesized late in the growth cycle. Although GNPS did not provide any annotations for these latter metabolites, query of multiple databases provided some clues as to their identities. For peak 1 there was 1 hit, a diketopiperazine with end absorption similar to what was observed. There were 3 possible hits (anthraquinones aspergilols A, B, and G) for peak 5 which included UV absorptions most similar to aspergilol G [28]; however, only by UV comparison, aspergilols A and B cannot be definitely excluded since they were also reported to have similar UV absorptions to peak 5. Two hits were found for peak 10 with similar UV profiles as the cyclopeptide epichloenin A and diketopiperazine-aluminum salt astalluminoxid (the former with reported antifungal activity) [29,30]. Peak 11 has a similar molecular weight to the antifungal Tyr-containing cyclic peptide KK-1 from the database Natural Product Atlas (record NPA028479), but their distinct UV properties rule out the possibility of such a match. [31].

Another four peaks (6, 9, 12, and 13) were all downregulated in experimental groups G5, G6, and G7. Peak 9 was nearly eliminated in all the groups, indicating that it may be not important to this fungal strains’ arsenal of allelochemicals. The down regulation of these four compounds is consistent with their not being involved in the antagonistic behavior of this fungus, and is consistent with economizing unused metabolic pathways during stress conditions. GNPS and database mining gave reliable annotations for peak 6 as the known AChE inhibitor territrem B [15,32,33] and for related peak 9 the antibacterial sesterterpene terretonin G. The latter annotation had similar UV characteristics [34]. For peak 12, five compounds with similar molecular weights and high polarity were found, i.e., Aspergillussanone H, Nigerasperone B, Aurasperone B, and Fumigatosides C-D from *Aspergillus* spp. [35,36,37,38,39,40], but their rich UV absorptive peaks ruled them out since peak 12 essentially only showed end absorption. Nevertheless, two possibilities, austamide (diketopiperizine) and asperimide A (containing a maleimide ring), had the most similar UV spectra to peak 13 out of the 3 hits [41,42]. Likewise, most of the *A. unguis* metabolites (peaks 3, 4, and 15) all were reduced in groups G6 and G7 when *A. unguis* was inoculated 1 week afterwards, reflecting the inhibition of *A. terreus* metabolites to *A. unguis* (Table 2). However, these three compounds changed in quite different ways in experimental groups G3, G4, and G5. For instance, peak 3 was produced in substantially higher yields in G5 (livU/livT, spontaneous inoculation) and G4 (livU-livT, *A. unguis* inoculated first) in contrast to its very low production level in G3 (iacU-livT). These results suggest that it was probably produced by *A. unguis* to oppose *A. terreus* and it was possibly a non-thermostable metabolite synthesized in its early growth stage. GNPS analysis along with a query of multiple databases did not provide a reasonable annotation based on molecular weight (776 Da), (MS^2^ features), and UV (end absorption) (Table 3).

Peak 4 was annotated as the cyclopeptide unguisin A by combining the multiple database mining [43], our previous work on this strain [17], and manual interpretation of its MS/MS spectrum (Figure 11). This compound was increased by more than 3 folds in *A. unguis* first inoculated groups G4 (livU-livT) and G3 (iacU-livT), whereas it was moderately downregulated in G5 (livU/livT). This profile suggests that it was useful in maintaining the predominance of *A. unguis* in co-culture, but is not likely a major component of its antagonistic arsenal.

Peak 15 showed a typical isotopic pattern for a tri-chlorinated compound by mass spectrometry (Table 3 and Appendix A), and was thus annotated as nidulin based on GNPS matching, multiple database mining [23], and our previous studies [17,18]. Its production in groups G3 (iacU-livT), G4 (livU-livT), and G5 (livU/livT) was reduced to different extents but not eliminated. In co-cultures G4 and G5 with live *A. unguis*, its yield still reached about 50% of the level observed in G1, suggesting that it played a positive yet minor role in this antagonistic response.

Noteworthily, peak 14 had an isotopic cluster typical of a dichlorinated metabolite (Table 3 and Appendix A) with four dichlorinated but UV distinct ‘hits’ in the kingdom of fungi, but no hits from the genus *Aspergillus* by GNPS matching or query of multiple databases [44,45]. Interestingly, it was not detected in monocultures of *A. unguis* and *A. terreus*. However, considering the observed halogenation ability of *A. unguis* [17,18,46], it is likely a metabolite of this latter fungus. It is intriguing that it was not produced in monoculture nor in G4 when it had the advantage of earlier inoculation, but only when the two strains were inoculated at the same time. As the competition was more intense as in G7 (livT-liv U) and *A. terreus* preemptively produced inhibitory factors as in G6 (iacT-liv U), its production was increased. This profile suggests that it might be a metabolite that provides resistance to the stress imposed by *A. terreus* metabolites. Furthermore, it is interesting that autoclaved *A. terreus* had a much stronger ability to induce production of peak 14 than live cultures, inferring that some potent and thermostable inducers are produced in the early stage of *A. terreus* growth.

Peak 2 is able to be produced by both *Aspergillus* species; however, the yield in *A. terreus* was about triple that of *A. unguis*; and under different co-culture conditions, its yields were higher than the monocultures; but it was not possible to discern which strain contributed more under the co-culture conditions. GNPS and multiple database mining did not give a clear indication of its identity with only one hit suggested, an anti-melanogenic triterpenoid saponin from *Aspergillus*. It had an identical molecular weight and end UV absorption [47]; however, more studies are necessary to confirm its identity and function as a potential allelochemical.

Briefly, the variation in features in different co-culture experiments revealed that both strains have allelochemicals that are produced in antagonistic conditions. Some metabolites appear to be involved in maintaining the predominance of first inoculated strain, others may be involved in anti-stress responses, and still others are down regulated in these conditions, presumably because they are not involved in these competitive interactions.

In addition to these 15 peaks observed in the LC-MS profiles, GNPS also annotated several other metabolites by MS/MS similarity comparisons as shown in Figure 9 and Figure 10. Some of these also varied in production levels in the different culture experiments, such as a siderophore-like desferrioxamine with *m/z* 478.228 together with its congener showing up in axenic *A. terreus* (G2) and other cultures in cluster F. Additionally, an open-ring di-chlorinated depsidone with *m/z* value 426.979 was produced in extremely low yield (Figure 10). However, classical GNPS networking can potentially sum the intensity of ions into the same node that possess ‘identical’ *m*/*z* values and MS/MS profiles but quite different retention times (e.g., isomers). Therefore, this quantification is not as reliable as EIC integration of the original LC-MS profiles.

## 3. Discussion

Based on the comparison of monocultures (axenic cultures) and co-cultures in different configurations for their mycelial morphology, HPTLC-bioautography analysis of secondary metabolic profiles, antimicrobial tests, and LC-PDA-MS/MS analysis, the first inoculated *Aspergillus* strains were commonly observed to predominate in co-cultures. This was true even when the culture was autoclaved one week into the experiment, as it still suppressed the growth and production of metabolites in the latter inoculated strain. When both strains of *Aspergillus* were inoculated simultaneously, they each grew in reasonable yield and produced their secondary metabolites (SMs). When *A. terreus* was inoculated before *A. unguis*, it produced its characteristic metabolites, including statins, undetermined peaks (e.g., 1, 5, 11), and desferrioxamines. Similarly, when *A. unguis* was first inoculated in the co-culture, it produced its typical metabolites, including unguisin A, nidulin, and undetermined peak 3. Meanwhile, the strains dramatically downregulated several metabolites that possessed no or only weak antimicrobial activity, and therefore, these latter compounds seem unrelated to the antagonistic phenotype of each fungus. This was especially the case when they were the latter inoculated strain in the co-cultural experiment. Nevertheless, *A. unguis* produced new dichlorinated metabolites only when it was extremely stressed by the simultaneous or second in the sequence inoculation, but not when it grew alone or was inoculated prior to *A. terreus*. On the other hand, *A. terreus* was induced to synthesize unknown compounds that did not ionize well in MS, but had strong white fluorescence on TLC analysis. This was especially the situation when simultaneously inoculated with *A. unguis*, which may be responsible for the enhanced antibacterial activity of extract of G5. In this case, the known antimicrobial agent nidulin was downregulated by *A. unguis* in this co-culture condition. Given this profile of expression, these unknown compounds may have anti-stress activities in the two *Aspergillus* strains. The overall results suggest that *A. unguis* was relatively ineffective in mounting an agonistic phenotype in culture, but was quite potent in stimulating its rival *A. terreus* to mount a strong response. The opposite was true for *A. terreus* in that it produced strongly antagonistic natural products but elicited little response from the competing fungus *A. unguis*.

Previous studies have revealed that co-cultivated microbes, such as fungi, often secrete extracellular diffusible SMs like phenols and quinones as well as enzymes such as phenoloxidases, peroxidases, and lignin-degrading enzymes, to suppress rivals or compete for new resources [48]. These competitive interactions occur even between different intraspecies strains; for example, the non-aflatoxigenic *Aspergillus flavus* can inhibit the aflatoxigenic *A. flavus* via antifungal SMs and antioxidants [49]. Numerous induced microbial SMs are involved in microbial antagonism as manifested by the fact that 37% have been reported with antibacterial, 7% with antifungal, and 35% with cytotoxic activities, together with 9% having other related activities including 3% as siderophores, 2% as α-glucosidase inhibitors, 1% as ATP synthesis inhibitors, and 3% as pesticides [50]. It has also been reported that the first inoculated strain in co-culture experiments usually predominates in the production of SMs, and that their yields can be even higher than in the corresponding mono-culture conditions. This concept was demonstrated by the co-cultivation of *Streptomyces rimosus* and *A. terreus* in a stirred tank bioreactor [51], and is reinforced by the findings of this present study.

Co-culture has increasingly been recognized as an efficient approach to activate silent biosynthetic gene clusters (BGC). This has been partially attributed to chromatin remodeling via epigenetic modifications [50,52]. For example, intimate physical contact of *Streptomyces rapamycinicus* to hyphae of *Aspergillus nidulans* has been shown to trigger the latter’s multi-subunit transcriptional co-activator complex SAGA/ADA, which in turns leads to histone H3K9 acetylation of transcription factor *basR* and *ors* BGC and the final production of SMs such as orsellinic acid [52,53]. Similar mechanisms are also known to occur in fungus-fungus co-cultures. The up-regulation of *O*-methylmellein was observed in the co-culture of two plant pathogens, *Eutypa lata* and *Botryosphaeria obtuse*. Interestingly, its production was also upregulated in the fungus *Stagonospora nodorum* by application of the epigenetic modifiers SAHA and nicotinamide [48,54,55]. Moreover, in a previous study by our group, a higher yield of unguisin A was observed in a *A. unguis* monoculture supplemented with both chemical epigenetic modifier procaine chloride and NaBr [17]. This previous finding is consistent with its significantly higher yields in the co-culture experiments G3 and G4 with *A. terreus*, and suggests that regulation of production of unguisin A may involve epigenetic mechanisms.

A literature-based search of MS-based features encountered in this study further supported their potential chemecological roles in these co-culture experiments. For example, unguisin A was initially reported from marine fungus *Emericella unguis* (the teleomorph of *A. unguis*); and while our previous investigations as well as a series of other studies did not detect potent antibacterial or antifungal activity for unguisin A [17,43,56], it was reported to be a promiscuous binder to various anions with particularly high affinity for phosphate and pyrophosphate [56]. Phosphorus is critical to fundamental life processes, and competition for phosphate is known to exist in microalgae-bacteria and microalgae-microalgae co-culture systems. Moreover, arbuscular mycorrhizal fungi have an important role in phosphorus absorption for their plant hosts [57,58,59]. Therefore, it is reasonable that in the *A. unguis*-*A. terreus* co-culture system of this present study, unguisin A may help *A. unguis* to acquire more phosphate and thus maintain its predominance over *A. terreus* when it is the first inoculated strain. It was surprising that in the G3 experiment (iacU-livT), this compound was maintained at high levels; this may be attributed to its relatively good stability. Additionally, it is conceivable *A. terreus* may use this *A. unguis*-derived compound to enhance its phosphate assimilation, as ’borrowing’ SMs from rival microbes is not uncommon [60].

Statin molecules such as lovastatin and simvastatin are typical *A. terreus* metabolites. They are used as lipid lowering drugs by inhibiting HMG-CoA in cholesterol biosynthesis, and have been shown to inhibit the growth of various *Aspergillus* spp. (*A. fumigatus**, A. flavus, A.niger, A. terreus,* etc.) and yeasts like *Candida albicans*. This growth inhibition occurs via multiple mechanisms, including HMG-CoA inhibition, iron starvation, induction of DNA fragmentation, fungal cell morphogenesis disruption, and others [27,61,62]. Variation in the production levels of statins was also investigated when *A. terreus* was co-cultivated with *Penicillium rubens*, *Chaetomium globosum,* and *Mucor racemosus*, respectively [63,64]. These experiments showed that the yields of lovastatin and its derivatives did not increase or were even lower than the yields in *A. terreus* monoculture, due partially to transformation into monacolin J. Nevertheless, in the present study the yields of lovastatin and its analog peak 8 were significantly larger compared to the monoculture; this may have been caused by the strong inducing effect of *A. unguis* towards *A. terreus*.

Halogenated compounds are an important class of marine fungal natural products (NPs). In 1994–2019, a total of 217 halogenated compounds were discovered from marine fungi, among which 88% were chlorinated compounds. Moreover, 18.9% were reported with antimicrobial activities, 35 were from the genus *Aspergillus* including 4 from *A. unguis* (all as depsidones), and none from *A. terreus* [65]. The PubChem database mainly records natural products from two original databases, the Natural Product Atlas (NPA) and the Natural Product Occurrence Database (NPOD). Duplicate records occurring in both databases are not excluded. Our search of PubChem returned 141 (from NPA) and 377 (from NPOD) *A. terreus* natural products, with 12 chlorinated ones (3 from NPA and 9 from NPOD) (total proportion 2.3%). From *A. unguis* we found that there were 19 NPs from NPA plus 58 NPs from NPOD with totally 24 (3 from NPA and 21 from NPOD) chlorinated ones (total proportion 31.2%). These data indicate that *A. unguis* possesses a much higher potential for producing chlorinated NPs than *A. terreus*. The tri-chlorinated NP nidulin that was found in this study is also known from *A. unguis* and *A. nidulans*, and has been characterized to possess antibacterial and moderate antifungal activities [17,18,64,65,66,67,68]. In co-cultural experiments G3-G6 in this study, the yield of nidulin was reduced although it remained at considerable levels of 14% to 56% of the yield from monocultures. We noticed that the extracts (1 mg/mL) of all the groups containing *A. unguis* that were growing well, such as axU (G1), showed comparable antimicrobial activity to the positive control ampicillin (0.1 mg/mL). This potent activity was possibly derived from nidulin, the main metabolite of *A. unguis*, and may also come from synergetic effects from other components. Furthermore, in experiments G5–G7, an unknown dichlorinated metabolite was upregulated under stress conditions. These results suggest that chlorinated NPs have chemical defensive functions in this fungus.

Siderophores are unique molecules that various organisms have evolved for the capture and assimilation of iron that is necessary for live. As such, they are intimately involved in iron resource competition during microbial co-cultures [51]. Hydroxamines (also as desferrioxamines, DFOs) are the most common siderophore family in nature and the main siderophore type produced by fungi. Fungi of the genus of *Aspergillus* are reported to produce DFOs consisting of ferrichromes and linear/cyclic fusarines [69,70]. The node with *m/z* value of 478.228 (Figure 9) was annotated by GNPS as a linear fusarine, although being different from any of the known *Aspergillus* fusarines. GNPS networking revealed that in the two monocultures, it was only produced by *A. terreus*. It was also present in the co-culture experiments G3, G5, and G7 where the live *A. terreus* was relatively easy to grow. These results are consistent with the use of this fusarine analog by *A. terreus* for iron uptake. These features along with the others discussed above may play important roles in the co-culture system.

In this current study, MS^2^-based molecular networking was employed based on data from an ion trap MS/MS instrument operating at low mass resolution. Using the GNPS platform, the combination of MS^1^ and MS^2^ data provided good putative annotations for metabolites observed in these co-culture experiments. However, for compounds unmatched by the GNPS analysis, a tedious manual searching across multiple NP databases was performed using likely molecular weights, taxonomical information, and UV spectral features. Nevertheless, solid verification of annotated or unknown metabolites requires confirmation by scaled-up fermentation, isolation, and structural elucidation by comprehensive spectroscopic methods. This confirmatory work is necessary and common to all mass spectrometry-based metabolomics studies, and is made especially challenging in co-culture studies which are inherently more difficult to reproducibly control.

In the current study, predominance of one fungus over the other in these co-culture experiments mainly relied on macroscopic morphological observations and comparisons of metabolite yields, neither of which method is direct or completely accurate. In future studies, microscopic observation, real-time quantitative PCR, microbiota amplifier high-throughput sequencing for ITS rDNA or metagenome sequencing, or transcriptomics approaches could be used to understand variations in the fungal community and gene expression more fully and accurately.

In summary, this study is the first report on the co-culture between *Aspergillus terreus* and *A. unguis* and reveals that *A. terreus* is more aggressive and responds to its rival’s presence, while *A. unguis* is less robust in its response but potent in stimulating its rival’s allelochemicals. It reveals their tendency to maintain dominance by synthesizing secondary metabolites and discloses the production of an unknown dichlorinated compound by *A. unguis* and strongly fluorescent products by *A. terreus* under stress. These results provide a deeper understanding of fungal co-culture mechanisms as well as the discovery of new natural products. This study also revealed a couple of shortcomings in the use of GNPS for compound annotation. The limited number of library records impacts the extent of automatic matching that can be achieved by GNPS. Additionally, an automatic analysis of isotopic peak clusters for recognizing the occurrence of halogenated compounds is current lacking. These aspects make necessary a complementary manual analysis of the data as well as querying of multiple databases via spectral features and taxonomic information.

## 4. Materials and Methods

### 4.1. Materials

The marine fungus A. *terreus* C23-3 was collected from the Xuwen Coral Reserve in Zhanjiang and is now preserved in the Guangdong Provincial Microbial Culture Collection, with the deposit number GDMCC No. 60316. The marine fungus A. *unguis* DLEP2008001 was derived from a red alga from the seaside of Fujiazhuang, Dalian, Liaoning Province, China and deposited in China General Microbiological Culture Collection Center with number CGMCC 3372. *Bacillus subtilis* MCCC 1A03710 was purchased from China Marine Microbial Culture Collection; *Pseudomonas aeruginosa* and *Candida albicans*, were from American Typical Culture Collection with numbers ATCC 9027 and ATCC 10231, respectively; *Vibrio parahaemolyticus* was donated by Professor Wen Chongqing, Fisheries College, Guangdong Ocean University; *V. alginolyticus* and *Shewanella putrefaciens* were presented by Professor Liu Ying from the School of Food Science and Technology, Guangdong Ocean University; *Methicillin-resistant Staphylococcus aureus* A7983 was gifted by Professor Yu Zhijun from Dalian Friendship Hospital. The acetylcholinesterase (AChE, from electric eels) and DPPH were purchased from Sigma-Aldrich (St. Louis, MO, USA). All the organic mobile phase solvents for LC-MS were from Merck (Darmstadt, Germany). All the other reagents were with analytical purity.

### 4.2. Methods

#### 4.2.1. Co-cultivation and Extraction of Strains

The strain A. *terreus* C23-3 and A. *unguis* DLEP2008001 were activated at 28 °C overnight and then inoculated, respectively, into conical culture flasks pre-filled with 200 mL of sterilized seawater potato solid medium for cultivation for 3-4 days until rich spores grew on the colonies. Then spore suspensions were prepared as seeds by washing the spores with sterile saline. Afterwards, the seed suspensions were inoculated into 500 mL Erlenmeyer flask each filled with 100 mL seawater potato sucrose liquid medium (containing 20 g sea salt, 20 g sucrose, and 500 mL potato juice per liter) according to the following group design. Each group contained three replicates.

Experiment grouping:
G1:*A. unguis* cultivated separately for 4 weeks (axenic *A. unguis*, abbreviated as axU);G2:*A. terreus* cultured separately for 4 weeks (axenic *A. terreus*, abbreviated as axT);G3:Inactivated *A. unguis* + live *A. terreus* (abbreviated as iacU-livT. In detail, *A. unguis* was inoculated first, cultured for one week and then iactivated by autoclaving; Afterwards, *A. terreus* was inoculated into the same flask and cultured for the next three weeks);G4:Live *A. unguis* + live *A. terreus* (abbreviated as livU-livT; Similar to G3, but the first inoculated *A. unguis* was not autoclaved);G5:Live *A. unguis*/Live *A. terreus* (abbreviated as livU/livT; The two strains were simultaneously inoculated into the same flask and cultivated for 4 weeks);G6:Inactivated *A. terreus* + live *A. unguis* (abbreviated as iacT/livU; Similar to G3, but the inoculation order was opposite);G7:Live *A. terreus* + *A. unguis* (abbreviated as livT-livU; Similar to G4, but the inoculation order was opposite).

During the 4-week cultivation, daily observation and photography were made to record the morphology of the cultures. After the four weeks, the fermentation broth was extracted three times with equal volumes of ethyl acetate. The mycelium was extracted three times with methanol assisted by ultrasonication; then the two extracts were concentrated, combined, and evaporated to dryness, weighed, and kept at 4 °C for further use.

#### 4.2.2. Thin Layer Chromatography (TLC) Analysis and Bioautography

All samples were prepared as methanol solution with concentration of 10 mg/mL of TLC analysis. The mobile phase was chloroform: methanol (volume ratio 20:1), the solid phase was Silica gel 60 F254 plate produced by Merck, and the volume was 10 μL. The plates were observed under 254 and 365 nm UV lights, stained by anisaldehyde-sulfuric acid reagent and potassium ferricyanide-ferric chloride (PFFC) reagent, respectively, or displayed for bioactive spots by DPPH free radical scavenging and AChE inhibitory bioautographies [71]. All the parallel samples were preliminarily checked for repeatability by TLC before formal experiments. All experimental results were recorded by photography.

#### 4.2.3. Antimicrobial Assay

The screening was performed using the bilayer agar plate-Oxford cup method [72,73]. The upper agar (5 mL) containing 0.5 mL of bacterial suspension with a concentration of 1 × 10^8^ CFU/mL. h = The Oxford cups (with inner diameter of 6 mm and outer diameter of 8 mm) were filled with 200 μL of sample (concentration = 1 mg/mL). The plates were incubated at 37 °C for 16–18 h. The results were expressed in average diameters of inhibition zones and standard deviations. The standard Muller–Hinton agar and Sabouraud agar were used for antibacterial and antifungal test, respectively (the medium of Vibrio parahaemolyticus contains 1% sodium chloride).

#### 4.2.4. LC-MS/MS Analysis

The nominal mass resolution LC–MS/MS analyses were run on a Thermo Finnigan LC-PDA-MS/MS system equipped with PDA Plus detector and a LCQ Advantage Plus ion-trap mass spectrometer (ThermoFinnigan, San Jose, CA, USA).

All the extract samples were prepared as methanol solutions in which the dryness contents were in proportion to the total crude extract amounts for each culture by weighting, dissolving with LC-MS pure methanol, and pretreating with Agilent SPE column. The DAD detector signal collection wavelength was 190–600 nm, and the monitoring wavelength was 210 nm, 254 nm, and 280 nm. The chromatographic column is a Phenomenex Kinetex C18 100A reverse-phase chromatographic column (100 × 4.60 mm, 5 μm). The detail of the mobile phase is listed in Table 4.

Mass spectrometry detection conditions: the mass scan range set to *m*/*z* 100–2000 Da, electrospray ionization; ion source: ion source voltage: 4 kV, capillary temperature: 325 °C, normalized collision energy: 35 eV, ion transfer tube voltage: 10 V. Trigger signal intensity threshold of the secondary mass spectrum: 1 × 10^5^ CPS (Count Per Second) for positive ion mode, and 1 × 10^4^ CPS for negative ion mode.

#### 4.2.5. GNPS Molecular Network Analysis

Standard pipeline for GNPS molecular networking was performed by referring to the previous reports [6,70]. The parameters for clustering and compound matching were set as: minimal matching fragments to be 4; minimal cluster size to be 2; cosine threshold to be 0.7; searching database scope to be the whole GNPS library. The data visualization is carried out with Cytoscape 3.7.2 software.

#### 4.2.6. Multiple Natural Product Databases Mining

The multiple natural product databases mining were performed on several open accessible online databases including the PubChem (https://pubchem.ncbi.nlm.nih.gov/#, 19 January 2022), the Dictionary of Natural Products (DNP) (http://dnp.chemnetbase.com/, 19 January 2022), the NPASS (http://bidd.group/NPASS/, 19 January 2022), the Natural Product Atlas (https://www.npatlas.org/, 19 January 2022), and the Nmrdata (WeiPu) (http://www.nmrdata.com/, 19 January 2022) using deduced molecular weights with error range of targeted MW ± 1 Da (or 2 Da for chlorinated metabolites), isotopic pattern (for chlorinated metabolites), UV features, and taxon information (mainly within the genus of *Aspergillus* and expanded to the kingdom of fungi when necessary).

## Figures and Tables

**Figure 1 antibiotics-11-00513-f001:**
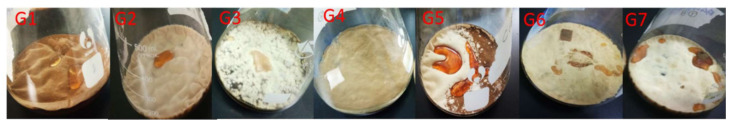
The morphology of axenic cultures and co-cultures in different experiments (28 days in total). (**G1**–**G7**), respectively, represent: axU (**G1**), axT (**G2**), iacU-livT (**G3**), livU-livT (**G4**), livU/livT (**G5**), iacT-livU (**G6**), livT-livU (**G7**).

**Figure 2 antibiotics-11-00513-f002:**
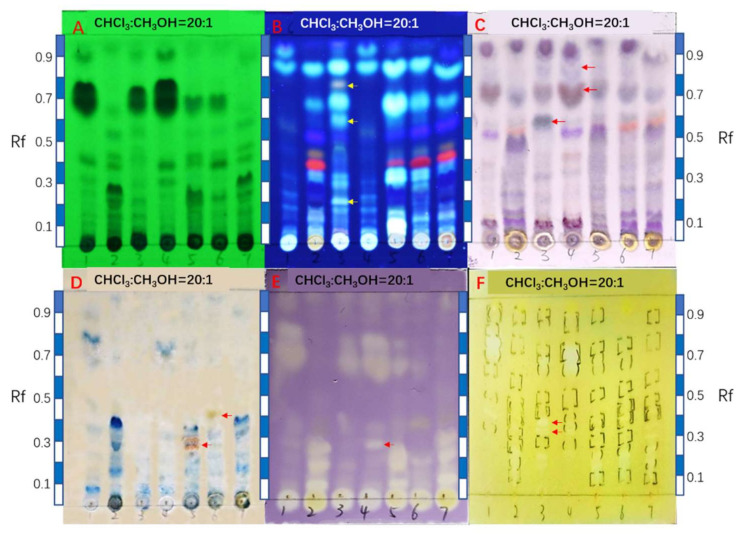
HPTLC fingerprints of the axenic and co-cultural extracts. (**A**) is the UV images of experiments G1–G7 under 254 nm (the sample numbers were marked with pencil below the starting line). (**B**) is the UV images of G1–G7 under 365 nm. (**C**) is the image of sulfuric acid-anisaldehyde colorized plate of G1–G7. (**D**) is the image of potassium ferricyanide-ferric chloride (PFFC) colorized plate of G1–G7. (**E**) is the DPPH free radical scavenging autographic image of G1–G7. (**F**) is acetylcholinesterase inhibitory bioautographic image of G1–G7. The developing agent was chloroform:methanol = 20:1 (*v*/*v*). The rulers beside the TLC plate are taken as references for Rf value calculation. The yellow or red arrows mark the new metabolites produced only under co-cultural conditions.

**Figure 3 antibiotics-11-00513-f003:**
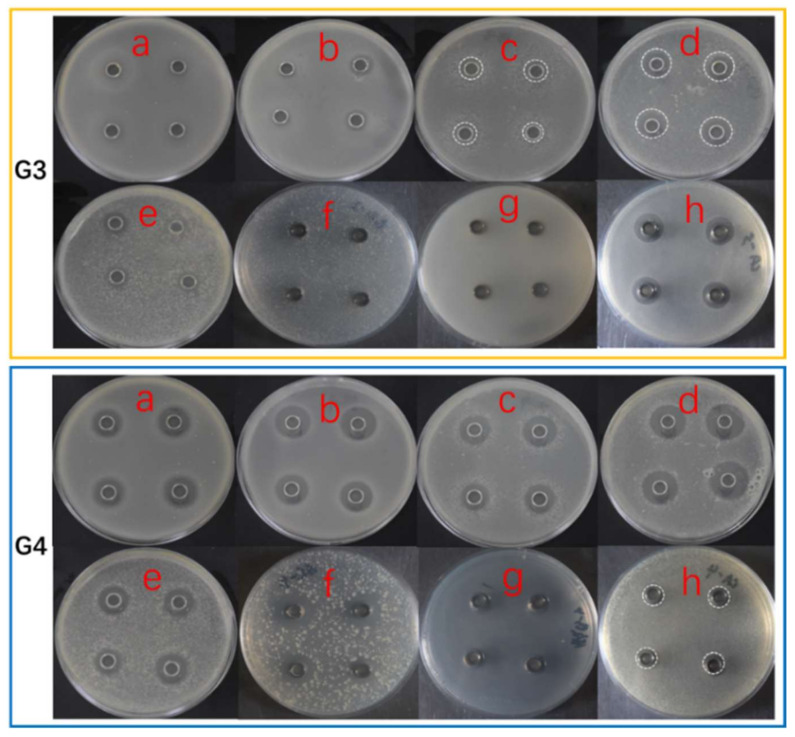
The antimicrobial activities of representative co-cultures (G3 & G4). The figures (**a**–**h**) for G3 & G4 are the antimicrobial results against MRSA, *Bacillus subtilis*, *Pseudomonas aeruginosa*, *Vibro parahemolyticus*, *V. alginolyticus*, *Shewanella putrefaciens*, *Yersinia pseudotuberculosis*, and *Candida albicans*, sequentially. For some relatively weak inhibition zones, circles in dash lines were used to mark them.

**Figure 4 antibiotics-11-00513-f004:**
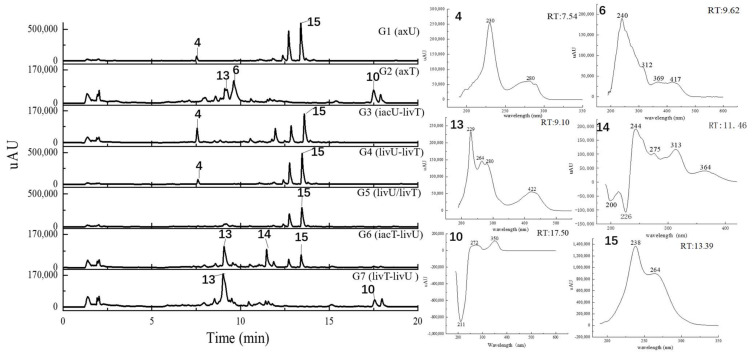
The HPLC traces of the culture extracts detected under the UV wavelength of 280 nm and the UV spectra for the featured peaks. The samples include axU (G1), axT (G2), iacU-livT (G3), livU-livT (G4), livU/livT (G5), iacT-livU (G6), and livT-livU (G7). The numbers marked on the peaks or in the UV spectra are numbers for the peaks with remarkable yield changes detected by mass spectrometry and are consistent with the peak numbers in Table 2.

**Figure 5 antibiotics-11-00513-f005:**
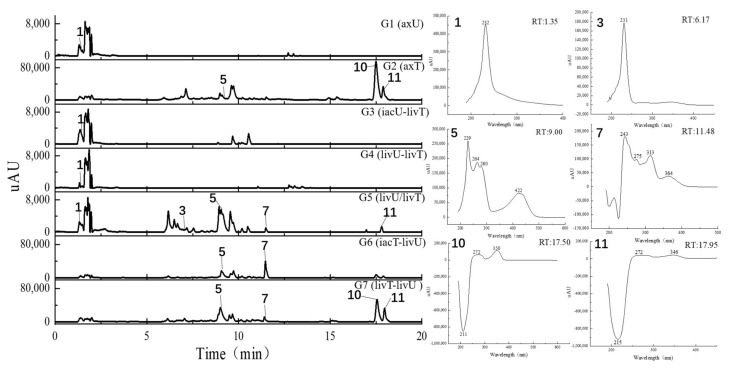
The HPLC traces of the culture extracts detected under the UV wavelength of 360–370 nm and the UV spectra for the featured peaks. The samples include axU (G1), axT (G2), iacU-livT (G3), livU-livT (G4), livU/livT (G5), iacT-livU (G6), and livT-livU (G7). The numbers marked on the peaks or in the UV spectra are numbers for the peaks with remarkable yield changes detected by mass spectrometry and are consistent with the peak numbers in Table 2.

**Figure 6 antibiotics-11-00513-f006:**
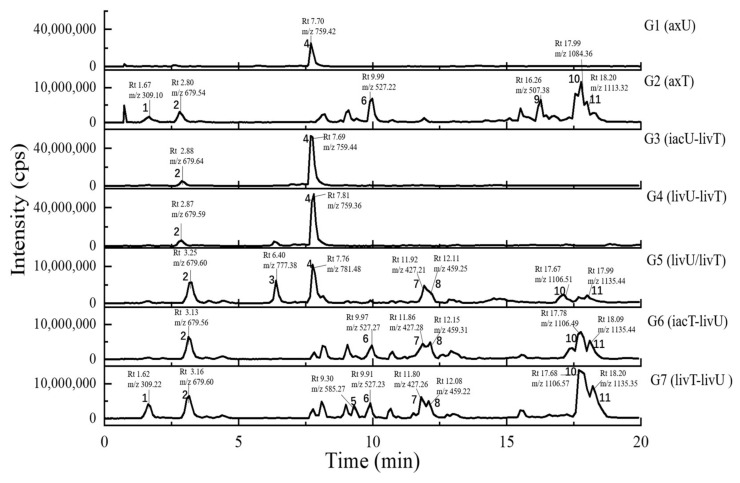
The LC-MS traces (base peak chromatographies, BPC) under positive ion mode of the culture extracts. The samples include axU (G1), axT (G2), iacU-livT (G3), livU-livT (G4), livU/livT (G5), iacT-livU (G6), and livT-livU (G7). The numbers marked on the peaks or in the UV spectra are numbers for the peaks with remarkable yield changes detected by mass spectrometry and are consistent with the peak numbers in Table 2.

**Figure 7 antibiotics-11-00513-f007:**
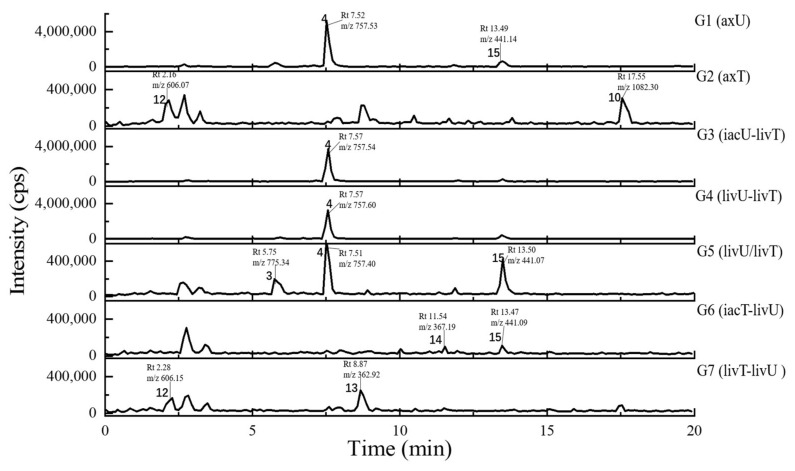
The LC-MS traces (base peak chromatographies, BPC) under negative ion mode of the culture extracts. The samples include axU (G1), axT (G2), iacU-livT (G3), livU-livT (G4), livU/livT (G5), iacT-livU (G6), and livT-livU (G7). The numbers marked on the peaks or in the UV spectra are numbers for the peaks with remarkable yield changes detected by mass spectrometry and are consistent with the peak numbers in Table 2.

**Figure 8 antibiotics-11-00513-f008:**
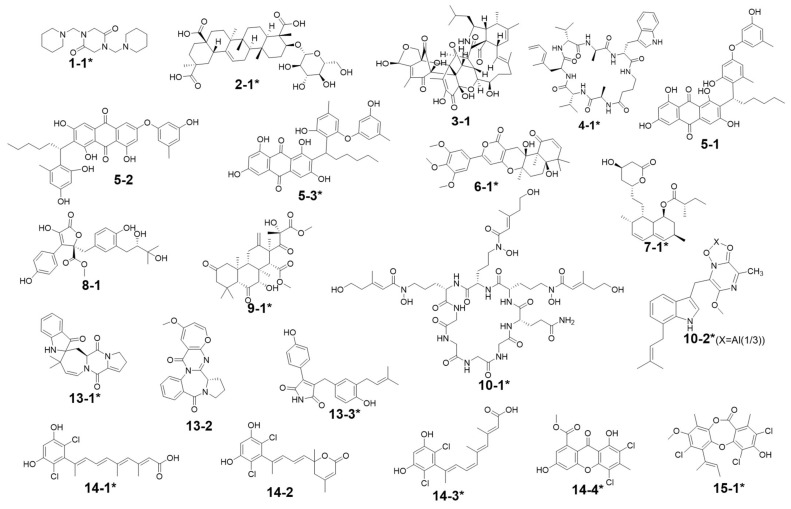
Annotated compound structures for the peaks 1–15. The asterisks (*) mark hits from databases with relatively higher reliability based on their similarities, including not only molecular weights, but also at least one of the following characteristics like GNPS MS^2^ similarity, UV features, isotopic patterns (for chlorinated compounds), and taxonomy, to the featured peaks. Compounds **10**-**1** and **10**-**2** represent two alternative annotations for peak 10 (see Table 3 for further data on each of these annotations). This terminology also applies for the other base compound numbers.

**Figure 9 antibiotics-11-00513-f009:**
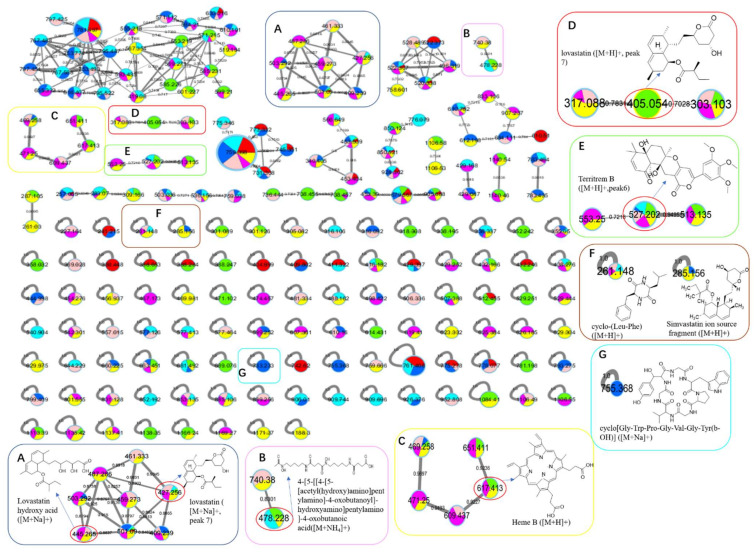
The GNPS molecular network based on positive ion MS/MS spectral similarity, showing a selection of amplified clusters. The nodes display the measured average masses of the molecular ions with identical MS/MS spectra. The sizes of the nodes reflect the relative amount of the corresponding compounds. The different colors of sections in the nodes represent different samples, i.e., 
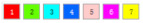
: axU (G1), axT (G2), iacU-livT (G3), livU-livT (G4), livU/livT (G5), iacT-livU (G6), and livT-livU (G7), respectively. (**A**) is an enlarged cluster for statins including the sodiated ion of peak 7 (lovastatin). (**B**) is an enlarged cluster containing a possible fusarine-like siderophore. (**C**) is an enlarged cluster containing heme B. (**D**) is an enlarged cluster containing protonated ion of peak 7. (**E**) is an enlarged cluster containing peak 6 (territrem B). (**F**) conclude two nodes annotated as a diketopiperazine and a simvastatin fragment. (**G**) is a node annotated as a cyclopeptide.

**Figure 10 antibiotics-11-00513-f010:**
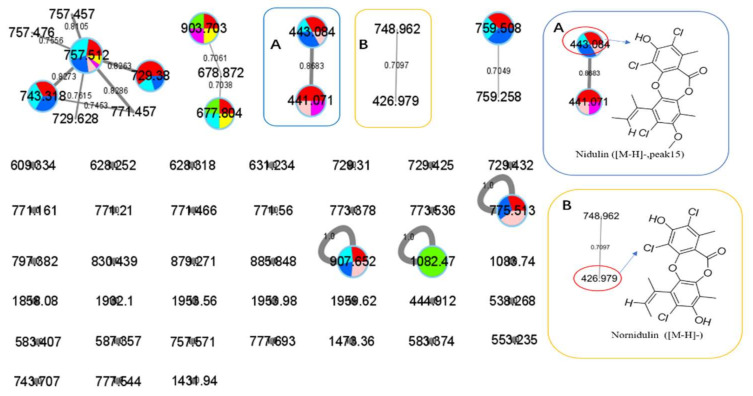
The GNPS molecular network based on negative ion MS/MS spectral similarity showing a selection of amplified clusters. The nodes display the measured average masses of the molecular ions with identical MS/MS spectra. The sizes of the nodes reflect the relative amount of the corresponding compounds. The different colors of sections in the nodes represent different samples, i.e., 
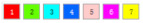
: axU (G1), axT (G2), iacU-livT (G3), livU-livT (G4), livU/livT (G5), iacT-livU (G6), and livT-livU (G7), respectively. (**A**) is an enlarged cluster containing peak 15 (nidulin). (**B**) is an enlarged cluster containing nornidulin.

**Figure 11 antibiotics-11-00513-f011:**
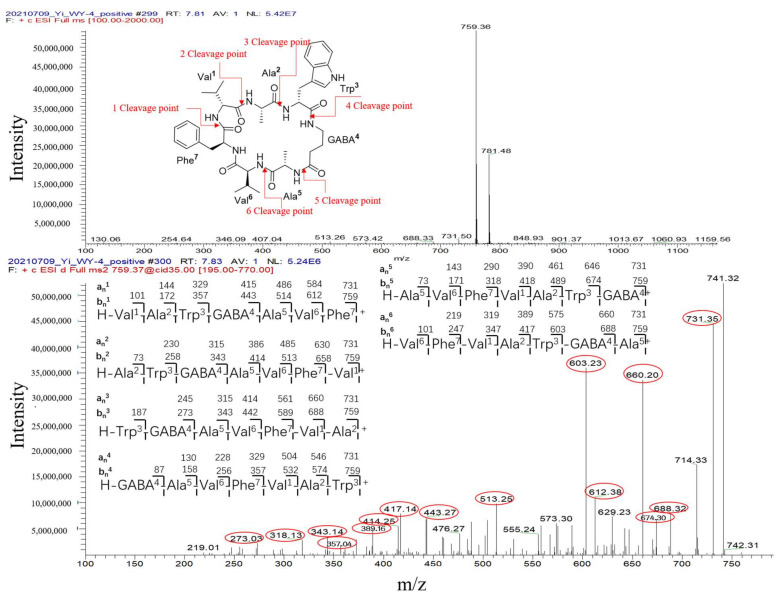
The interpretation of MS/MS spectrum of peak 4 (cyclopeptide unguisin A).

**Table 1 antibiotics-11-00513-t001:** The total extract amounts and antimicrobial activities from experiments G1–G7, which were measured using the Oxford Cup method (dosage: 200 mL/well, concentration = 1 mg/mL, concentration for both controls = 0.1 mg/mL, *n* = 4).

Sample Number	Total Sample Amount (Yield: mg/flask)	Diameters of Inhibition Zones Against Indicator Microbes (mm) ^#^							
		MRSA	*Bacillus subtilis*	*Pseudomonus* *aeruginosa*	*Vibro* *parahemolyticus*	*Vibro* *alginolyticus*	*Shewanella* *putrefaciens*	*Yersinia* *pseudotuberculosis*	*Candida* *albicans*
axU (G1)	345 ± 40	15.7 ± 0.6	13.2 ± 0.5	14.0 ± 0.9	18.6 ± 0.9	17.8 ± 0.8	17.1 ± 1.2	-	13 ± 0.5
axT (G2)	624 ± 10	-	-	-	7.3 ± 0.4	8.1 ± 0.6	9.3 ± 0.5	-	-
iacU-livT (G3)	560 ± 20	-	7.5 ± 0.8	8.9 ± 0.9	11.8 ± 0.4	7.3 ± 0.9	7.1 ± 0.5	-	14.2 ± 0.7
livU-livT (G4)	309 ± 30	13.0 ± 0.4	15.7 ± 0.5	15.6 ± 1.0	17.8 ± 0.6	16.2 ± 0.6	17.4 ± 0.5	-	11.7 ± 0.5
livU/livT (G5)	420 ± 80	10.2 ± 0.6	14.3 ± 0.7	14.2 ± 0.7	-	14.3 ± 0.5	18 ± 0.8	-	13.0 ± 0.8
iacT-livU (G6)	440 ± 20	-	7.0 ± 0.4	-	-	9.6 ± 0.6	10.4 ± 0.8	-	12.3 ± 0.5
livT-livU (G7)	638 ± 40	8.1 ± 0.3	-	-	8. 1 ± 0.9	-	-	-	9.2 ± 0.7
Ampicillin	-	14.1 ± 0.4	17.5 ± 0.2	19.7 ± 0.5	24.1 ± 0.9	18.6 ± 0.4	16.1 ± 0.6	11.3 ± 0.2	-
Ketoconazole	-	-	-	-	-	-	-	-	16.3 ± 0.8

^#^: three times average ± standard deviation; -: no activity or very weak activity.

**Table 2 antibiotics-11-00513-t002:** The changing folds for the yields of the differential peaks.

	Feature Peak Number	G3(iacU-livT)	G4(livU-livT)	G5(livU/livT)	G6(iacT-livU)	G7(livT-livU)
G2(axT)	1	**↓**0.14	**↓**<0.01	**↓**0.02	**↓**0.13	↑1.58
	2	↑1.59	↑1.96	↑2.11	↑2.24	↑2.46
	5	↓0.23	**↓**0.02	**↓**0.08	↑1.49	↑3.49
	6	**↓**0.12	**↓**<0.01	**↓**0.11	↓0.7	↓0.72
	7	↓0.21	**↓**0.08	↑3.57	↑3.15	↑4.42
	8	↓0.52	**↓**0.19	**↑**8.41	**↑**7.31	**↑**10.47
	9	**↓**0.02	**↓**0.06	**↓**<0.01	**↓**0.08	**↓**0.05
	10	↓<0.01	↓<0.01	**↓**0.09	↓0.64	↑1.36
	11	**↓**0.02	**↓**0.04	↓0.52	↑1.33	↑3.03
	12	↓0.2	**↓**0.06	**↓**0.08	**↓**0.02	↓0.57
	13	**↓**0.03	**↓**0.02	**↓**0.14	↓0.27	↓0.91
G1(axU)	2	↑4.57	**↑**5.66	**↑**6.07	**↑**6.45	**↑**7.09
	3	**↓**0.05	↑3.46	↑3.08	**↓**<0.01	**↓**0.01
	4	↑3.65	↑3.48	↓0.78	**↓**0.09	**↓**0.14
	15	↓0.23	↓0.56	↓0.5	**↓**0.14	**↓**0
	14	**↓**0	**↓**0	↑1	**↑**13.39	**↑**1.89

Note: **↓**: Production decreased fivefold and more; ↓: Production decreased less than fivefold; **↑**: Production increased fivefold and more; ↑: Production increased less than fivefold; **↑**: New metabolite (in large quantities) in co-culture; ↑: New metabolite (in small amounts) in co-culture.

**Table 3 antibiotics-11-00513-t003:** Multiple database mining of the main peaks that show remarkably different yields in base peak chromatographies (BPCs) of their LC-MS profiles using positive and negative modes of ionization.

Peak Number	Presence in Sample	*m*/*z* Value Measured	Retention Time (min)	UV Maximum Measured (nm)	Compound Hits in Library	Molecular Weight in Libraries	Libraries & IDs	MS^2^ Similarity (Cosine)	Molecular Formula	UV Maximum Absportive Peaks in Libraries/Literature (nm)	Bioresource	DOI	Biological Activity	Structures Code of the Compound Hits
1	G2, G7	309.10 [M + H]^+^ (presumed)	1.67	232	1,4-Bis(piperidin-1-ylmethyl)piperazine-2,5-dione	308.33	Dictionary of Natural Products	N/A	C_16_H_28_N_4_O_2_	N/A (isolate amide)	*A. terreus*	N/A	N/A	**1-1 ***
2	G1–G7	679.59 [M + H]^+^ 677.88 [M − H]^−^	2.87	231	3β-(β-D-glucopyranosyloxy)olean-12-ene-23,28,30-trioic acid	678.81	NMRDATA, 1331571; Natural Product Atlas, NPA026397; Pubchem, 146682840	N/A	C_36_H_54_O_12_	N/A (isolate double bonds)	*A. amstelodami*	10.1002/cbdv.201900237	anti-melanogenic and anti-allergic activity	**2-1 ***
3	G1, G4, G5	777.38 [M + H]^+^ 775.34 [M − H]^−^	6.40	231	Aspergilasine B	775.84	NMRDATA, 999923; Dictionary of Natural Products	N/A	C_42_H_49_NO_13_	202, 240	*A. flavipes QCS12*	10.1021/acs.orglett.7b02146	no inhibitory activities againest seven cancer cell lines up to a concentration of 40μM.	**3-1**
4	G1, G3–G7	759.36 [M + H]^+^ 757.60 [M − H]^−^	7.81	230, 280	Unguisin A	758.92	NMRDATA, 29553	N/A	C_40_H_54_N_8_O_7_	290, 281, 274, 219	*A. unguis*	10.1021/np980539z;10.1039/C7OB00316A	moderately inhibited *Staphylococcus aureus*;as an anion receptor with high affinity for phosphate and pyrophosphate	**4-1 ***
5	G2, G7	585.27 [M + H]^+^ 583.24 [M − H]^−^	9.30	229, 264, 280, 422	Aspergilol A	584.62	NMRDATA, 895659; Pubchem, 132915662; Natural Product Atlas, NPA009011; Dictionary of Natural Products	N/A	C_34_H_32_O_9_	196, 293, 452	*A. versicolor*	10.1016/j.tet.2015.10.038	possessing antioxidant activities	**5-1**
					Aspergilol B	584.61	Dictionary of Natural Products	N/A	C_34_H_32_O_9_	194, 293, 462	*Aspergillus*	10.1016/j.tet.2015.10.038	possessing antioxidant activities	**5-2**
					Aspergilol G	584.61	Dictionary of Natural Products	N/A	C_34_H_32_O_9_	206, 265, 295, 458	*Aspergillus*	10.1016/J.BMCL.2017.01.032	N/A	**5-3 ***
6	G2, G6, G7	527.22 [M + H]^+^	9.99	240, 312, 369, 417	Territrem B	526.57	GNPS, CCMSLIB00005436075	0.70	C_29_H_34_O_9_	195, 220, 236, 330, 284	*A. terreus*	10.3390/md12126113	strong inhibitory activity against acetylcholinesterase, potent antifouling activity	**6-1 ***
						526.57	Dictionary of Natural Products	N/A						
7	G2, G5–G7	427.26 [M + Na]^+^	11.80	243, 275, 313, 364	Lovastatin	404.54	GNPS, CCMSLIB00000852214	0.76	C_24_H_36_O_5_	231, 238, 247	*A. terreus*	10.1080/10826068.2020.1805624	the competitive inhibitors of the enzyme hydroxy-methyl-glutaryl coenzyme A (HMG-CoA) reductase	**7-1 ***
						404.54	Dictionary of Natural Products	N/A						
8	G5–G7	459.31 [M + H]^+^ (presumed)	12.15	246, 285	Unannotated statin	N/A	N/A	N/A	N/A	N/A	N/A	N/A	N/A	N/A
					Aspernolide D	458.46	NMRDATA, 152713; Pubchem, 46930025; Natural Product Atlas, NPA003511	N/A	C_24_H_26_O_9_	290	*A. terreus RCBC1002*	10.1248/cpb.58.1221	Inactive against all bacterial strains	**8-1**
9	G2, G7	507.38 [M + H]^+^ (presumed)	16.26	N/A (no obvious absorption)	Terretonin G	506.58	NMRDATA, 809567; Dictionary of Natural Products	N/A	C_27_H_38_O_9_	End absorption	*Aspergillus* sp. *OPMF00272*	10.1038/ja.2014.46	Moderate antimicrobial activity against Gram-positive bacteria	**9-1 ***
10	G2, G5–G7	1106.49 [M + Na]^+^ 1082.38 [M − H]^−^	17.78	272, 350	Epichloenin A	1083.15	Dictionary of Natural Products	N/A	C_46_H_74_N_12_O_18_	N/A (containing a,b-unsaturated amides)	*Epichloe¨ festucae*	10.1371/journal.ppat.1003332	as an important molecular/cellular signal for controlling fungal growth and hence the symbiotic interaction.	**10-1 ***
					Astalluminoxide	1084.22	Natural Product Atlas, NPA032177	N/A	C_60_H_66_AlN_9_O_9_	201, 222, 272, 349	*A. terreus BCC51799*	10.1016/j.tet.2020.131496	moderate to weak cytotoxicity against both cancerous and non-cancerous cells.	**10-2 ***
11	G2, G5–G7	1135.35 [M + Na]^+^	18.20	273, 346	N/A	N/A	N/A	N/A	N/A	N/A	N/A	N/A	N/A	N/A
12	G2, G7	606.07 [M − H]^−^ (presumed)	2.16	231	N/A	N/A	N/A	N/A	N/A	N/A	N/A	N/A	N/A	N/A
13	G2, G7	362.92 [M − H]^−^ (presumed)	8.87	226, 264, 278, 420	Austamide	363.41	Dictionary of Natural Products	N/A	C_21_H_21_N_3_O_3_	234, 256, 282, 392	*A.ustus*	10.1016/s0040-4039(01)97170-9	toxic to ducklings	**13-1 ***
					Circumdatin B	363.37	Dictionary of Natural Products	N/A	C_20_H_17_N_3_O_4_	284, 358	*A. ochraceus*	10.1021/jo981536u	Inactive in the assay against NCI’s 60 cancer cell line panel	**13-2**
					Asperimide A	363.41	Natural Product Atlas, NPA028229	N/A	C_22_H_21_NO_4_	229, 278, 360	*A.terreus*	10.1016/j.fitote.2018.10.011	not found exhibited cytotoxicity	**13-3 ***
14	G6, G7	367.19 [M − H]^−^ (presumed) (367.19:369.11:371.19 = 9:6:1, in intensity) revealing to be dichlorinated compound	11.54	244, 275, 313, 361	Cosmochlorin A	369.24	Dictionary of Natural Products; Natural Product Atlas, NPA030107	N/A	C_18_H_18_Cl_2_O_4_	323	*Cosmospora vilior* IM2-155	10.1016/j.phytol.2016.09.007	moderate antimicrobial activity against gram-positive bacteria and fungi; partially restored the growth inhibition caused by hyperactivated Ca^2+^-signaling in mutant yeast and showed GSK-3b inhibition	**14-1 ***
					Cosmochlorin B	369.24	Dictionary of Natural Products; Natural Product Atlas, NPA030108	N/A	C_18_H_18_Cl_2_O_4_	230, 290	*Cosmospora vilior* IM2-155	10.1016/j.phytol.2016.09.007	Inactive against microbes; similar restoring the growth inhibition activity to cosmochlorin A, promoting osteoclast formation	**14-2 ***
					Cosmochlorin C	369.24	Dictionary of Natural Products; Natural Product Atlas, NPA030109	N/A	C_18_H_18_Cl_2_O_4_	323	*Cosmospora vilior* IM2-155	10.1016/j.phytol.2016.09.007	Similar antimicrobial activity to cosmochlorin A	**14-3 ***
					Penicillixanthone	369.15	Dictionary of Natural Products; Natural Product Atlas, NPA008373	N/A	C_16_H_10_Cl_2_O_6_	230, 294, 369	*Penicillium* sp. PSU-RSPG99	10.1016/j.tet.2014.05.105	No antimycobacterial and cytotoxic activities	**14-4 ***
15	G1, G3–G6	441.07 [M − H]^−^ (441.07:443.02:444.99:446.12 = 27:27:9:1, in intensity)	13.50	237, 264	Nidulin	443.70	GNPS, CCMSLIB00005436077	0.70	C_20_H_17_Cl_3_O_5_	267	*A. unguis*	10.1055/s-0031-1298228 10.1080/14786419.2013.879305	aromatase inhibitory and antimicrobial and DNA damaging activities	**15-1 ***
							Pubchem, 6450195; Dictionary of Natural Products	N/A						

N/A: indicates not applicable or with no record or having limited accessibility. Compounds can be found by name in the Dictionary of Natural Product online database with more details and ID numbers. The asterisks (*) mark hits from databases with relatively higher reliability based on their similarities, including not only molecular weights, but also at least one of the following characteristics, like GNPS MS^2^ similarity, UV features, isotopic patterns (for chlorinated compounds), and taxonomy, to the featured peaks.

**Table 4 antibiotics-11-00513-t004:** Chromatographic analysis conditions.

Injection Volume (μL)	Elution Conditions		Flow Rate (mL/min)
	Time (min)	Proportion	
25	0.00–1.00	30% ACN-H_2_O	0.6
	1.00–10.00	30–99% ACN-H_2_O	
	10.00–16.00	99% ACN-H_2_O	
	16.00–16.20	99–30% ACN-H_2_O	
	16.20–20.00	30% ACN-H_2_O	

Note: The Mobile phase contained 0.1% formic acid.

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
