# Peer review of "Secondary Metabolite Variation and Bioactivities of Two Marine Aspergillus Strains in Static Co-Culture Investigated by Molecular Network Analysis and Multiple Database Mining Based on LC-PDA-MS/MS"

_antibiotics, 2022, doi:10.3390/antibiotics11040513_

Round 1

Reviewer 1 Report

Manuscript can be accepted

Author Response

Reviewer: 1

Manuscript can be accepted

Response: Thanks for your kind suggestion. To improve our manuscript, we have made careful checking for minor errors and modification according to other reviewers’ suggestion mainly about the simplifying of the table and structures figure of the hits during revision. They were shown by the track mode of WORD.

We hope our revisions are adequate. Together with this Response Letter, we prepare two files for the same revised version. In one of them, all track changes and annotations are clearly shown; In the other, we hide the tracks and only show the annotations to make the layout clean. If any further question, please contact us.

Thank you for your nice comment again!

Sincerely yours,

Yi Zhang

Reviewer 2 Report

This is an interesting paper, worth publishing, but a few minor issues should be considered and/or corrected:

Lines 230-234: those data could be easier to read if shaped in a table format.

“Compounds 1-1, 1-2, 1-3, and 1-4 represent four annotations for peak 1 (see Table 3 for further data on each of these  annotations).” This sentence is rather confusing, as it is not clear whether the four annotations are alternative annotations corresponding to a single pure peak or simultaneous annotations corresponding to an impure peak (a mixture of four compounds). It is assumed that the authors have checked the peak purity and could clarify this aspect so as to avoid any equivocal meaning of the sentence.

Section 4.2.3. There is no information on the two controls used in the antimicrobial assessment (their names are available in Table 1, but concentrations used are not provided).

The discussions section should also include at least one paragraph on the significance and relevance of the antimicrobials results reported in Table 1 (for instance, the fact that axU (G1) seems superior to ampicillin in its action on MRSA, particularly considering that this is a mixture, but we do not know whether the concentrations were equivalent, so as to allow a direct relevant comparison).

Author Response

Reviewer: 2

  1. Lines 230-234: those data could be easier to read if shaped in a table format.

Response: We have modified lines 230-234 of these chromatographic conditions into tabular form.

Table 4. Chromatographic analysis conditions.

Injection volume (μL)

Elution conditions

Flow rate (mL/min)

Time (min)

Proportion

25

0.00-1.00

30% ACN-H2O

0.6

1.00-10.00

30%-99% ACN-H2O

10.00-16.00

99% ACN-H2O

16.00-16.20

99%-30% ACN-H2O

16.20-20.00

30% ACN-H2O

 Note: the Mobile phase contained 0.1% formic acid.

  1. “Compounds 1-1, 1-2, 1-3, and 1-4 represent four annotations for peak 1 (see Table 3 for further data on each of these annotations).” This sentence is rather confusing, as it is not clear whether the four annotations are alternative annotations corresponding to a single pure peak or simultaneous annotations corresponding to an impure peak (a mixture of four compounds). It is assumed that the authors have checked the peak purity and could clarify this aspect so as to avoid any equivocal meaning of the sentence.

Response: Thank you for your reminder. Yes, compounds 1-1, 1-2, 1-3 and 1-4 represent the four alternative annotations for peak 1. Now, we add ‘alternative’ before the word ‘annotation’.

  1. Section 4.2.3. There is no information on the two controls used in the antimicrobial assessment (their names are available in Table 1, but concentrations used are not provided).

Response: The concentrations for the two controls were both 0.1 mg/mL. We now indicate this in the title of Table 1 clearly.

  1. The discussions section should also include at least one paragraph on the significance and relevance of the antimicrobials results reported in Table 1 (for instance, the fact that axU (G1) seems superior to ampicillin in its action on MRSA, particularly considering that this is a mixture, but we do not know whether the concentrations were equivalent, so as to allow a direct relevant comparison).

Response: As we replied in our answer to the above third question to reviewer 2 and newly supplemented in Table 1, the concentration of crude extracts were ten fold of the positive controls. And considering that Aspergillus unguis has nidulin as a main metabolite with potent antimicrobial activity and synergetic effects from other components, it is reasonable that the crude extracts like axU(G1) that have a much higher concentration than ampicillin showed comparable activity potency to the latter. Now, we have also added this to the sixth paragraph about halogenated compounds including nidulin in the discussion section. Additionally, in the discussion section like the first paragraph in the previous version, we also had some discussion about the antimicrobial activity.

Besides the reviewers’ suggestions, we also corrected some minor errors when we were checking the manuscript. They were shown by the track mode of WORD.

We hope our revisions are adequate. Together with this Response Letter, we prepare two files for the same revised version. In one of them, all track changes and annotations are clearly shown; In the other, we hide the tracks and only show the annotations to make the layout clean. If any further question, please contact us. Thank you all for your strict attitude and kind suggestions for the improvement of our manuscript.

Sincerely yours,

Yi Zhang

Reviewer 3 Report

Please see the attached file containing comments and suggestions for the authors.

Author Response

Reviewer: 3

  1. Page 1, line 43: "effect t." Remove the extra "t" and space. line 45: "assisted by" instead of "aided with" would be better English. Page 2, line 96: "studiees".Page 3, line 108: "…aromatase), and diphenyl-picryl…". Page 5 line 223: "G1-G7".

Response: We are sorry for the previous carelessness. We have improved the language according to these questions you mentioned, see details in red color and the related annotations in the revised version of the manuscript.

  1. Page 5, line 207, Figure 2 caption, line 224 and Page 23 (= 2nd Page 4), line 206 (? Why do the Page numbers and line counts restart on Page 20 so we have Pages 1-19 of 28 then Pages 1-9 of 28? )

Response: The previous discontinuity of line numbers and page number may have been caused by the improper insertion of figures. Now, we have resolved this problem in the revised version of the manuscript.

  1. There is a discrepancy here between the solvent ratio stated (20:1) here and that indicated at the top of all plates 2A-2F (10:1) in Figure 2.

Response: In Figure 2, the solvent ratio of all figures 2A-2F should be 20:1. The new figure with corrected heading marks has been inserted into the revised version of the manuscript.

  1. Page 12, Table 2: A space or a line is needed to indicate where G2 finishes and G1 starts; alternately: move G2(axT) to the first line of data, and align G2(axU) with the second feature peak 2 (after peak 13).

Response: Now, we have added a line in Table 2 to indicate where G2 finishes and G1 starts, and have moved G2(axT) to the first line of data, and align G2(axU) with the second feature peak 2 (after peak 13) according to the reviewer’s suggestion.

  1. Page 13, Figure 9: Diagrams A and D are missing an OH substituent on the lovastatin lactone ring, and depict a wedge bond at the sp2 planar ring junction.

Response: In the revised version of the manuscript, figure 9 has been corrected for the structure of lovastatin in Diagrams A and D. In figure 8, this error has also been corrected.

  1. Page 14, lines 381-383:This sentence isn’t Engilish, and I have no idea what it is trying to convery.

Response: Did the reviewer mean ‘Peak 11 has similar properties to the antifungal cyclopeptide KK-1 for peak 11, have their distinct UV features excludes this possibility as an exact match’? Now, we have corrected this sentence as ‘Peak 11 has similar mass to the antifungal Tyr-containing cyclic peptide KK-1 from the database Natural Product Atlas (NPA028479), but their distinct UV properties rule out the possibility of such a match.’, please see the change in the text and the related annotation in the revised version of the manuscript.

  1. Page 15, line 392: 7 hits , but there are 8 compounds listed under peak 12 in Table 3 and only 6 structures are included in Figure 2 (12-1 to 12-6)?

Response: For this question and the following similar questions (8-10 and 12-14) about the selection of the hits, we have compared in detail to exclude the previous hits with obviously inconsistent UV absorptions or isotopic patterns to the correspondent peaks. In fact, we had also considered this problem previously and used asterisks to rank the reliability according to their different characteristics [GNPS MS2 similarity, UV features, isotopic patterns (for chlorinated compounds), and/or taxonomy]. However, previously we conservatively kept more candidates as in Table 3 and Figure 8 for the reason that deleting too many seemed risky and less convincing. But the reviewer’s suggestion is reasonable that we should make the results simplified and clearer. So, now we have tried our best to delete the redundant hits. However, for some potential hits with somewhat similar UV spectra, we have kept them in the list, recognizing that moderate changes in reported absorption versus our observations could occur as a result of differences in solvents and experimental conditions. Now, on basis of greatly reduced number of hits, we continue to use the previous while simplified asterisks (*) in Figure 8 and in Table 3 to mark the hits from databases with relatively higher reliability based on their similarities, including not only molecular weights, but also at least one of the following characteristics: GNPS MS2 similarity, UV features, isotopic patterns (for chlorinated compounds), and taxonomy, to the featured peaks. We also explained this selection in the caption of Figure 8 and the footnote of Table 3.

     For peak 12, we have deleted all the previous hits from Table 3 and Figure 8 because they showed quite different and richly detailed UV absorptive maximums compared with peak 12 that essentially showed only end absorption. But we concisely cited five previous ‘hits’ with close molecular weights to peak 12 and explained why we did not take them as hits in the previous paragraph. Please see details in the text, in Figure 8, Table 3 and the related annotations.

  1. Page 11, Figure 8 is quite crowded and rather messy, with overlapping within structures (e.g. 3-1, 3-2, 4-2, 5-1 to 5-3, 7-3, etc.), between adjacent structures and significant distortion of some structures (e.g. the indole ring in 3-2, a wedge bond at a planar sp2 ring junction in 7-1 as well as 180o/60obond angles at an sp2 centre in 12.1)

Response: As explained above, we have deleted many redundant structures. Meanwhile, we have also greatly improved the layout so as to avoide crowding and overlap. Significant bond distortion of some structures has been corrected.

  1. Specifically, for starters,of the 4"annotated" compound structures for peak 1, your observed UV data effectively rules out structures 1-1, 1-2 and 1-4 as potential annotations as they are all highly conjugated structures and would have UV absorptions at much longer wavelengths  than 232nm, while 1-3, an amide, would be the only candidate you have identified with the right  molecular weight that would be expected to absorb near the observed frequency.

Response: As explained above in response to question 7, we have deleted redundant structures including the previous 1-1, 1-2 and 1-4.

  1. For peak 3, you indicate two annotated structures , but only one of those structures (Asperversiamide A, 3-2), has  the correct molecular weight that corresponds to that observed (776, based on [M+H]+ and [M-H]- peaks at 777 and 775 respectively). Aspergilasine B (M. Wt 775), on the other hand, should give  [M+H]+ and [M-H]- peaks at 776 and 774 respectively. So, why is Aspergilasine B an annotated  structure when it has the wrong mass?

Response: As explained above in response to question 7, we have deleted redundant structures including Asperversiamide A (3-2) due to its obviously inconsistent UV spectra to peak 3 (end absorption). However, we kept Aspergilasine B as a possible hit because of its relatively close UV spectra or at least not easily excludable possibility of UV similarity to peak 3. As for the problem of mass difference, on account of the relatively broad mass error range (+/- 1 Da) when we use data from nominal resolution mass spectrometer, Aspergilasine B (M. Wt 775) should not be definitely ruled out from the deduced molecular weight range on basis of [M+H]+ and [M-H]- peaks at 777 and 775, respectively. That means measured m/z 777 could actually be 776-778 and 775 could be actually 774-776. So, when we did database searching, we used target MW ± 1 Da (for peaks 14 and 15 containing chlorinated metabolites, we used target MW ± 2 Da) as mass range in case missing possible candidates. Now, we also add this detail concerning criterion into the Method section 4.2.6.

  1. All charges in the column should be superscripts. (At present, that is only the case for peak 7). I think there also is an issue with most, if not all, of the Molecular Weight in Libraries values in Table 3. What you need to use for comparison are the calculated single predominant isotope molecular weights (because they are what the MS is measuring). It appears that the library values are probably all calculated values weighted using natura.

Response: We corrected all the charges on the column. When we previously searched the databases, we used a broad mass range of target MW ± 1(or 2) Da not only in consideration of measurement errors from our nominal mass spectrometer, but also in consideration of covering the small difference between average and exact mass. So, this concern will not have a significant influence or impact on our results.

  1. For peak 4, the UV data (230, 280nm) is consistent with 4-1 (amide + indole) but doesn't exclude the presence of 4-2 (amide only), however, the molecular weights of the structures again differ by 1 amu, and only Unguisin A, 4-1, M.Wt 758 would afford [M+H]+ and [M-H]~ peaks at 759 and 757 respectively: Aspercryptin A1 would have its peaks at 760 and 758, so isn't compatible with the experimental data, so why is it an“annotated structure"?

Response: As explained above in response to question 7, we have deleted redundancy structures including aspercryptin A1 since its UV did not fit peak 7 well. As for molecular weights, unguisin A  (average mass: 758.92 Da; exact mass: 758.4115) and aspercryptin A1 (average mass: 758.01 Da; exact mass: 757.5313) are both in the reasonable range of measured [M+H]+ (759.36) and [M-H]- (757.60) for peak 4 taking the mass error of nominal mass spectrometer into account. Finally, we keep unguisin A as the hit for peak 12 because of not only MW and UV, but also the interpretation of MS2 spectra and the basis of previous study for this A. unguis strain as we had previously described in the paragraph for it in Results section.

  1. Aspergilols A, B and G, 5-1 to 5-3 all contain the same chromophores and are isomeric, so would only potentially be distinguished by retention times and/or their MS/MS fragmentation patterns. There doesn't seem to be any discussion on how 5.3 was deduced as the likely“annotated structure".

Response: For this question, now we state as ‘There were 3 possible hits (anthraquinones aspergilols A, B, and G) for peak 5 which included UV absorptions most similar to aspergilol G [27]; however, only by UV comparison, aspergilols A and B cannot be definitely excluded since they were also reported to have similar UV absorptions to peak 5’ in the text. So, we have kept aspergilos A and B in Table 3 and Figure 8.

  1. For peak 6, 2“annotated structures" , 6-1 and 6-2 are indicated, BUT, one contains CI and the other doesn't, so why is it that the expected 3:1 isotope ratios for chlorine isotopes 2 mass units apart hasn't been used to rule out one of the structures? The observed MS data should clearly tell you whether CI is or isn't present, and rule out one of the“annotated structures".

Response: As explained above in response to question 7, now we have deleted redundancy structures including the previous ‘6-2’ due to the helpful reason suggested by the reviewer.

  1. Table 3 require significant revision and appropriate discussion that explains the choice of asterisked (and double asterisked) structures for all peaks of interest needs to be added.

Response: As explained above in response to question 7, now we have deleted redundant structures from Table 3 and Figure 8. For all asterisked structures, we have explained about this in the Results section. Additionally, we have simplified the way of asterisk marking without using ‘**’.

Besides the reviewers’ suggestions, we also corrected some minor errors when we were checking the manuscript. They were shown by the track mode of WORD.

We hope our revisions are adequate. Together with this Response Letter, we prepare two files for the same revised version. In one of them, all track changes and annotations are clearly shown; In the other, we hide the tracks and only show the annotations to make the layout clean. If any further question, please contact us. Thank you all for your strict attitude and kind suggestions for the improvement of our manuscript.

Sincerely yours,

Yi Zhang

Round 2

Reviewer 3 Report

 I am satisfied with the responses from the authors and with the alterations they have made, so am giving the go-ahead for publication (despite the fact that I think that low resolution MS/MSMS should be quite capable of distinguishing 1 amu for atomic masses of <1000: i.e. < 1000ppm differences, even in complex mixtures!)